# Recognition and reporting of suspected adverse drug reactions by surveyed healthcare professionals in Uganda: key determinants

Ronald Kiguba,[1] Charles Karamagi,[2] Paul Waako,[1] Helen B Ndagije,[3] Sheila M Bird[4]

▶ Prepublication history and additional material is available. To view please visit the journal (http://dx.doi.org/10.1136/bmjopen-2014-005869).

For numbered affiliations see end of article.

Correspondence to
Professor Sheila M Bird;
sheila.bird@mrc-bsu.cam.ac.uk

## ABSTRACT

**Objective:** To assess extent and determinants of past-month recognition of suspected adverse drug reactions (ADR) and past-year ADR reporting among healthcare professionals (HCPs) in Uganda.

**Setting:** Geographically diverse health facilities (public, private for-profit, private not-for-profit).

**Participants:** Of 2000 questionnaires distributed, 1345 were completed: return rate of 67%.

**Primary and secondary outcome measures:** Per cent HCPs who suspected ADR in the past month; reported ADR in the past year.

**Results:** Nurses were the majority (59%, 792/1345). Only half the respondents had heard about pharmacovigilance: 39% of nurses (295/763; 95% CI 35% to 42%), 70% otherwise (383/547; 95% CI 66% to 74%). One fifth (268/1289 or 21%; 95% CI 19% to 23%) had suspected an ADR in the previous 4 weeks, 111 of them were nurses; 15% (190/1296) had reported a suspected ADR in the past year, 103 of them were nurses. Past-month ADR suspicion was more likely by non-nurses (OR=1.7, 95% CI 1.16 to 2.40) and with medical research involvement (OR=1.5, 95% CI 1.05 to 2.15) but past-month receipt of patient ADR-complaint predominated (OR=19, 95% CI 14 to 28). Past-year ADR reporting was higher by hospital staff (OR=1.9, 95% CI 1.18 to 3.10), especially in medicine (OR=2.3, 95% CI 1.08 to 4.73); but lower from private for-profit health facilities (OR=0.5, 95% CI 0.28 to 0.77) and by older staff (OR=0.6, 95% CI 0.43 to 0.91); more likely by HCPs who had ever encountered a fatal ADR (OR=2.9, 95% CI 1.94 to 4.25), knew to whom to report (OR=1.7, 95% CI 1.18 to 2.46), or suggested how to improve ADR reporting (OR=1.6, 95% CI 1.04 to 2.49). Two attitudinal factors were important: diffidence and lethargy.

**Conclusions:** One in five HCPs suspected an ADR in the past-month and one in seven reported ADR in the previous year. Empowering patients could strengthen ADR detection and reporting in Africa.

## BACKGROUND

Adverse drug reactions (ADRs) are significant causes of patient morbidity and mortality[1]

### Strengths and limitations of this study

- Over 1300 healthcare professionals surveyed in diverse health facilities in Uganda.
- Attitudes to pharmacovigilance elicited.
- Demographic and professional determinants ascertained of past-month adverse drug reaction (ADR) suspicion and past-year ADR reporting.
- Purposely selected survey locations and non-random sampling of healthcare professionals.
- Under-representation of nurses.

and are known to raise overall healthcare costs.[2–5] The WHO[6] defines pharmacovigilance (PV) as "the science and activities relating to the detection, assessment, understanding and prevention of adverse effects or any other medicine-related problem." Spontaneous and voluntary reporting of suspected ADRs generates signals about rare, delayed and unexpected drug reactions that are undetected in the initial phases of drug development[7] under-reporting is a major limitation[8] of all ADRs are reported.[9–11] This low rate of ADR reporting undermines efforts to identify and estimate the magnitude of drug risks, confirmation of actionable issues and possible regulatory action.[12]

Widespread use of electronic medical record databases has enhanced patient safety through automation of signal detections for ADRs, thereby improving healthcare service delivery.[13] In Africa, the establishment and use of such databases is still rare[14] and ADR reporting is largely done manually. Strengthening of PV systems in sub-Saharan African (SSA) countries has received support from global health initiatives, but reporting is often disease specific (eg, malaria, vaccines, HIV/AIDS) because of restricted funding streams rather than strengthening

countrywide reporting systems.[15] As a result, PV systems in SSA remain weak.[16] In Uganda, 556 spontaneous reports were submitted to the National Pharmacovigilance Centre (NPC) in the initial 5 years of 2005–2009. Of these, 315 (57%) were related to medicines with 10 or more spontaneous ADR reports and were dominated by antiretroviral drugs (51%, 160/315), antimalarials (27%, 85/315) and antibiotics (22%, 70/315).[17] The dominance of ADR reports related to these groups of medicines accords with the burden of disease in SSA.[18]

The WHO's Uppsala Monitoring Centre (UMS) maintains web-based ADR reporting software (VigiFlow) for use by NPCs.[19] Although receipt of 200 or more ADR reports per million population per year is desirable,[20] most SSA countries submitted fewer than 20 ADR reports per million population in 2010 compared to more than 100 reports per million in other low-income and middle-income countries.[21]

Uganda established a NPC in 2005 and has been a member of the WHO programme for International Drug Monitoring since 2007. In 2010, there was a training-of-trainers session for 30 national PV trainers. By 2011, 14 regional PV centres were established;[21] PV-training sessions for core teams of healthcare professionals (HCPs) were conducted in each of these centres and ADR reporting forms distributed.[22] At least one support supervision visit per centre is conducted annually. Despite these efforts, before the reporting rate in Uganda (population: 36 million) is still low at 6 ADR reports per million population per year, based on 1348 ADR reports in 2007–2012 (180, 75, 229,[23] 140, 183, 413 in 2012 (when Targeted Spontaneous Reporting (TSR) was launched); and 128 in January–June 2013 (Nassali Huldah and Helen Ndagije, personal communication, 15 January 2014)). Moreover, significant missing information in four-fifths of ADR reports compromises analysis.[17]

Of 46 SSA countries from which PV systems were assessed to determine their capacity to ensure drug safety, Uganda was identified as one of four with active PV systems that could, in principle, detect, evaluate and address medicine safety issues.[24] Indeed, Ugandan surgical series[25] on, and subsequent media coverage of, gluteal fibrosis and post-injection paralysis among children injected with quinine[26 27] triggered investigation by the Ugandan NPC, which, in 2010, mediated change of Uganda's recommended quinine injection site from the gluteus muscle to the thigh.[28]

Personal and professional characteristics associated with increased ADR reporting by HCPs include older age, male, lower workload, higher number of prescriptions issued per day, type of education received, specific PV training and involvement in teaching and research.[8 29 30] Inhibitory factors include: unavailability of ADR forms, bureaucratic method of ADR reporting and uncertainty over which professional cadre is mandated to report ADRs.[31]

In 1996, Inman et al[32] described eight 'deadly sins' to explain why HCPs under-report ADRs: (1) attitudes related to professional activities (financial incentives, fear of litigation and ambition to publish personal case series), (2) ADR-related knowledge and attitudes (complacency, diffidence, indifference and ignorance) and (3) excuses made by HCPs (lethargy). Insecurity is an attitudinal factor that was not proposed by Inman but has been reported elsewhere.[33]

In Africa, there is a paucity of empirical data on PV awareness.[34–38] Hence we sought to determine the level of PV awareness by HCPs, the extent and determinants of past-month ADR recognition and of past-year ADR reporting in Uganda.

## METHODS
### Study design and sampling procedure
From 25 May 2012 through 28 February 2013, we conducted a survey across Uganda in purposively selected, geographically diverse public and private health facilities. Public institutions included the National Referral Hospital-Mulago, and six regional referral hospitals each selected to represent a major region of the country. In addition, we included district hospitals and health centres (HCs) at levels II to IV in the catchment area where a Regional Referral Hospital was selected. For logistical reasons, we selected a convenience sample of private for-profit and private not-for-profit health facilities (which included drug shops) in the respective districts where public institutions were assessed. Permission to conduct the research was sought from the administrators of the selected institutions.

Any HCP involved in prescribing, transcribing, dispensing medication orders and administration of drugs to a patient was eligible for inclusion. Written informed consent was obtained from HCPs prior to their recruitment. The self-completed questionnaires did not contain identifying information on individual HCPs. The survey team used serial numbers to track distributed questionnaires. Five research assistants, all final year medical students at Mulago National Referral Hospital, were initially recruited and trained on the concepts of PV, informed consent, response rate and on the survey questionnaire, which they self-completed. Completion of questionnaire by research assistants was primarily to familiarise them with it and to gauge time to completion (22, 25, 27, 31 and 31; mean of 27 min) but served also as a brief pretest. A similar model of data collection by pretrained investigators was employed in the upcountry sites.

Given the challenge of accessing staff lists in the selected health facilities (and especially so in private for-profit settings), random sampling of eligible HCPs was not practicable. Instead, in each health facility, the pretrained investigators approached HCPs of all ranks and invited them to complete a pretested questionnaire, of which 2200 were printed and 2000 distributed.

Invitations might be declined if HCPs were particularly busy or, despite willingness, a delay of several days or weeks might ensue before the self-completion questionnaire was returned. In practice, neither the refusal rate by approached HCPs nor the 'did not return rate', by professional cadre, for distributed questionnaires was reliably documented.

In Uganda, there were reckoned to be 46 566 HCPs in 2009,[39] who would have been survey-eligible had they worked at the survey-locations. Doctors and dentists (3459) represented an estimated 7% of the nationally eligible staff but were 20% of the achieved sample; 762 pharmacists and pharmacy technicians 1.6% of nationally eligible staff but 6% of the achieved sample; and 37 625 nurses, midwives and nursing assistants an estimated 81% of the nationally eligible staff but 59% of the achieved sample.

### Data collection and management

The survey questionnaire, see online supplementary appendix, elicited demographic and professional information, description of the most recent suspected ADR and attitudes to, as well as knowledge and use of, the suspected ADR reporting system. The questionnaire for HCPs included 15 attitudinal statements on ADR reporting, which were scored from 1 (total disagreement) to 5 (total agreement). All data were entered into a databank using EpiData V.3.1.

Prior to its administration, the questionnaire was elaborated between members of the research team who have diverse expertise in pharmacy, PV and questionnaire design. Completion-time was tested by research assistants. Thereafter, an integrated pilot study was conducted on 125 HCPs. The subsequent revisions were sufficiently minor so that results of the pretest were included in the final analysis.

### Statistical analysis

Responses are summarised as frequencies and percentages. Different potential determinants for the past-month recognition or past-year reporting of suspected ADRs were screened using $\chi^2$ tests for categorical variables. Logistic regression was then used to assess the relationship of demographic and professional factors in several ways, eg: (i) recognition of suspected ADRs in the past 4 weeks; and for those in post for at least 1 year and (ii) having reported at least one suspected ADR in the past 12 months. Attitudinal factors were also incorporated in (ii). Missing data were accounted for using the missing assigned approach where infrequently missing data were meaningfully assigned to an existing category (rather than by multiple imputations under the missing at random assumption).[40] Results are expressed as ORs with 95% CIs. Statistical analyses were carried out using Stata V.12.0.[41]

## RESULTS

### Study population

Of 2000 questionnaires distributed, 1345 were completed, a return rate of 67%. Mean age of respondent HCPs was 32.4 years (SD=8.9). Nurses were the majority (792/1345 or 59%), see table 1.

### Awareness of pharmacovigilance

Only half the respondents (678/1310 or 52%; 95% CI 49% to 55%) had ever heard about PV: two-fifths of nurses (295/763 or 39%; 95% CI 35% to 42%) and 70% of others (383/547; 95% CI 66% to 74%). Thirty per cent of HCPs (412/1317; 95% CI 29% to 34%) were aware of the existence of Uganda's NPC but only 3% (37/1312; 95% CI 2% to 4%) of HCPs had *ever* submitted an ADR report to the NPC.

### Suspected ADR reporting in the previous 12 months

Only 15% of HCPs (190/1296; 95% CI 13% to 17%) had reported a suspected ADR in the previous 12 months, of whom 15% (27/175) claimed to have made their report to NPC so that our respondents' past-year ADR reporting rate to NPC was an estimated 1 in 50 (2%). Only 41% (11/27; 95% CI 22% to 59%) past-

**Table 1** Demographic and professional characteristics of healthcare professionals, Uganda, 2013

| Total number of participants | 1345 |
| --- | --- |
| Age, n=1253 | |
| Mean years (SD); median, IQR | 32.4 (8.9); 30, 26–36 |
| Gender, n=1345 | |
| Male | 541 (40.2) |
| Female | 804 (59.8) |
| Number of patients seen per day, n=1226 | |
| Mean number (SD); median, IQR | 41.0 (46.3); 30, 15–50 |
| Professional cadre, n=1345 | |
| Nurse | 792 (58.9) |
| Doctor | 275 (20.4) |
| Pharmacist and pharmacy technician | 84 (6.3) |
| Other | 194 (14.4) |
| Type of health facility, n=1345 | |
| Public | 568 (42.2) |
| Private not-for-profit | 280 (20.8) |
| Private for-profit | 497 (37.0) |
| Highest academic qualification, n=1345 | |
| Certificate | 471 (35.0) |
| Diploma | 501 (37.3) |
| First degree | 294 (21.9) |
| Masters degree or PhD | 79 (5.9) |
| Ever received ADR training, n=1225 | |
| Yes | 180 (14.7) |
| No | 1045 (85.3) |
| Received patient ADR complaint in past 4 weeks, n=1302 | |
| Yes | 340 (26.1) |
| No | 962 (73.9) |

ADR, adverse drug reaction.

year reporters to NPC had found the NPC form clear on what to report.

When HCPs were asked about when, in the past 12 months, they had reported their most recent suspected ADR, 79/178 (44%) said within the past month, 28 (16%) in the months 2+3 prior, and 71 (40%) in months 4–12, a distribution indicative either of a multiplicity of reports per ADR reporter or biased recall.

## ADR recognition

Twenty-one per cent (268/1289: 95% CI 19% to 23%) of respondents had suspected an ADR in the previous 1 month, 76% of whom (195/257: 95% CI 71% to 81%) had received patient ADR-complaints in the past month. Of HCPs who had suspected an ADR in the past month, 35% (92/262: 95% CI 29% to 41%) had reported an ADR in the past 12 months.

Among HCPs who had not suspected an ADR in the previous month, 12% (121/1000: 95% CI 10% to 14%) had nonetheless received patient ADR-complaints in the past month.

In the previous 4 weeks, see table 2, 26% (340/1302) of HCPs had received 1190 patient ADR-complaints (mean of 3.5 complaints (SD 9.5) per complaint-receiving HCP) which equates to 0.9 ADR-complaints (95% CI 0.65 to 1.18) per HCP per month. Also, 21% (268/1289) of HCPs had suspected 670 ADRs (mean of 2.5 suspected ADRs (SD 2.6) per suspecting HCP), which equates to 0.5 suspected ADRs (95% CI 0.45 to 0.59) per HCP per month, implying an ADR suspicion rate of 0.57 (0.52/0.91) per patient ADR-complaint per HCP per month (95% CI 0.42 to 0.80).

Among the 15% (190/1296) who were ADR reporters in the previous 12 months, 44% (79/178) claimed to have submitted their most recent report in the past 4 weeks. If so, there could be at least 84 suspected ADR reports submitted by 1296 HCPs in the past 4 weeks (or 0.065 ADR-reports in past 4 weeks per HCP) when 0.5 ADRs were suspected in the past 4 weeks per HCP. This translates into a 13% ADR-report rate per suspected ADR.

## Medication classes and fatalities in survey-described suspected ADRs

The most frequently mentioned medication classes associated with 182 survey-described ADRs in the past 4 weeks that cited one or more drugs (216 drug citations) were antibiotics (38%, 83/216), antiretroviral agents (23%, 49/216), antimalarials (15%, 33/216, 15 of which implicated quinine), analgaesics (9%, 19/216) and others (15%, 32/216).

Two suspected ADRs were described by HCPs and involved child fatalities in association with quinine: a 5–year-old girl had been given intravenous quinine and died soon after arrival at a private not-for-profit hospital in Eastern Uganda; and a 2-year old boy had reacted to quinine and died despite the doctor in a public hospital in Eastern Uganda having administered an antidote. Full

**Table 2** Patient ADR-complaints and HCPs' ADR suspicion in past 4 weeks, Uganda, 2013

Patient ADR-complaints/HCPs' ADR suspicion

Patient ADR-complaints in past 4 weeks

| Cadre | Number of HCPs | Who received complaints | Mean (SD) ADR-complaints | ADR-complaints received | ADR-complaints per HCP |
|---|---|---|---|---|---|
| Overall | 1302 | 340 (26%) | 3.5 (9.5) | 1190 | 0.91 |
| Nurses | 760 | 155 (20%) | 3.9 (11.4) | 604 | 0.80 |
| Non-nurses | 542 | 185 (34%) | 3.2 (7.7) | 592 | 1.09 |
| Doctors | 270 | 97 (36%) | 3.3 (10.2) | 320 | 1.19 |
| Pharm/Ptech | 81 | 34 (42%) | 3.9 (4.0) | 132 | 1.64 |
| Other | 191 | 54 (28%) | 2.5 (2.1) | 135 | 0.71 |

HCPs' ADR suspicion in past 4 weeks

| Cadre | Number of HCPs | Who suspected ADRs | Mean (SD) suspected ADRs | ADR suspicions by HCPs | ADR suspicion per HCP |
|---|---|---|---|---|---|
| Overall | 1289 | 268 (21%) | 2.5 (2.6) | 670 | 0.52 |
| Nurses | 756 | 111 (15%) | 2.6 (2.6) | 288 | 0.38 |
| Non-nurses | 533 | 157 (29%) | 2.5 (2.6) | 393 | 0.74 |
| Doctors | 267 | 88 (33%) | 2.3 (2.5) | 202 | 0.76 |
| Pharm/Ptech | 80 | 23 (29%) | 2.9 (3.2) | 66 | 0.83 |
| Other | 186 | 46 (25%) | 2.5 (2.5) | 114 | 0.61 |

ADR, adverse drug reaction; HCP, healthcare professional; Pharm/Ptech, pharmacist and pharmacy technician.

details of HCPs describing suspected ADRs will be reported separately.

### Feedback to ADR reporters

Reporters of ADRs to AIDS Treatment Information Centre (ATIC) received the highest feedback (60%, 12/20), followed by those who reported to the Medical Superintendent or Institutional Review Board (39%: 23/58+4/11). Feedback from Uganda's NPC was infrequent (23%:5/22). Reporters of ADRs to drug manufacturers (4) or District Directors of Health Services (12) received zero feedback.

### Reasons for ADR reporting

The commonest reason that respondents vouched for ADR reporting was that the patient had developed a serious ADR (30%, 48/159 reasons) followed by patient safety (18%, 29/159) and patient ADR-complaint (8%, 13/159). The next three reasons each had nine citations: institutional mandate to report ADRs, prevention of similar ADRs and as a means of obtaining advice.

### Attitudes to ADR reporting

Only 14% (186/1301:95% CI 12% to 16%) of respondents indicated that reporting ADRs put their career at risk, see table 3, while 36% (466/1304:95% CI 33% to 38%) thought that it is only necessary to report serious or unexpected ADRs. Most respondents agreed that they have a professional obligation to report ADRs (76%, 1000/1311:95% CI 74% to 79%) and 68% (896/1319:95% CI 65% to 70%) stated that they would report ADRs if there were an easier method. Forty-five per cent (596/1312:95% CI 43% to 48%) stated that they do not know how information reported in the ADR form is used, 64% (833/1309:95% CI 61% to 66%) felt that they would report an ADR only if they were sure it was related to use of a particular drug and 27% (349/1305: 95% CI 24% to 29%) felt that they should be financially reimbursed for providing the ADR reporting service.

### Factors associated with ADR suspicion in the past month

Suspicion of ADR in the past 4 weeks was more likely by non-nurses (OR=1.7, 95% CI 1.16 to 2.40) and with involvement in medical research (OR=1.5, 95% CI 1.05 to 2.15), but the clearly dominant factor was that the HCP had received patient ADR-complaint(s) in the past 4 weeks (OR=19, 95% CI 14 to 28). There was some evidence that ADR suspicion was less likely by staff in surgical wards, see table 4.

Logistic regression analysis among the 973 respondents who did not receive a patient ADR-complaint did not identify any additional significant cofactors associated with ADR suspicion.

### Personal, professional and attitudinal factors associated with having made an ADR report in the past 12 months

Demographic and professional factors associated with a lower likelihood to report ADRs in the past 12 months were: private for-profit health facility (vs public; OR=0.5, 95% CI 0.28 to 0.77) and HCP aged over 30 years (OR=0.6, 95% CI 0.43 to 0.91); while those associated with being more likely to report ADRs included: medical department (vs surgery; OR=2.3, 95% CI 1.08 to 4.73), having ever encountered a fatal ADR (OR=2.9, 95% CI 1.94 to 4.25), knowing to whom to report ADRs (OR=1.7, 95% CI 1.18 to 2.46) and HCPs who had suggested ways of improved ADR reporting (OR=1.6, 95% CI 1.04 to 2.49), see table 5.

Only two attitudinal factors were additionally relevant: diffidence ('the belief that reporting an ADR would only

**Table 3** Healthcare professionals' responses to 15 attitudinal statements on adverse drug reaction (ADR) reporting, Uganda, 2013

| Statement | Agree | Neutral | Disagree |
|---|---|---|---|
| Serious ADRs are well documented by the time a drug is marketed | 820 (61.7) | 166 (12.5) | 343 (25.8) |
| It is nearly impossible to determine whether a drug is responsible for a particular adverse reaction | 527 (39.8) | 189 (14.3) | 607 (45.9) |
| I would only report an ADR if I were sure that it was related to the use of a particular drug | 833 (63.6) | 138 (10.6) | 338 (25.8) |
| The one case of an ADR that an individual health worker might see makes no significant contribution to medical knowledge | 210 (16.2) | 122 (9.4) | 966 (74.4) |
| I read articles about adverse drug reactions with interest | 824 (63.3) | 180 (13.8) | 298 (22.9) |
| I have a professional obligation to report ADRs | 1000 (76.3) | 143 (10.9) | 168 (12.8) |
| Reporting ADRs puts my career at risk | 186 (14.3) | 126 (9.7) | 989 (76.0) |
| It is only necessary to report serious or unexpected ADRs | 466 (35.7) | 129 (9.9) | 709 (54.4) |
| I do not have time to complete an ADR report form | 143 (10.9) | 208 (15.8) | 963 (73.3) |
| I do not have the time to actively look for ADRs while at work | 195 (14.8) | 152 (11.6) | 968 (73.6) |
| I do not know how information reported in an ADR form is used | 596 (45.4) | 194 (14.8) | 522 (39.8) |
| I talk with pharmaceutical companies about possible ADRs with their drugs | 290 (22.2) | 202 (15.5) | 813 (62.3) |
| I think the best way to report ADRs is by publishing in medical literature | 701 (53.4) | 238 (18.1) | 374 (28.5) |
| I should be financially reimbursed for providing the ADR service | 349 (26.7) | 199 (15.3) | 757 (58.0) |
| I would be more likely to report ADRs if there were an easier method | 896 (67.9) | 169 (12.8) | 254 (19.3) |

**Table 4** Personal and professional factors associated with ADR suspicion in the past 4 weeks among 1289 healthcare professionals, Uganda, 2013

| Factor | ADR suspicion | | Crude analysis | | | Adjusted analysis | | |
|---|---|---|---|---|---|---|---|---|
| | Yes (%) | No (%) | OR | 95% CI | p Value | OR | 95% CI | p Value |
| **Level of health facility** | | | | | | | | |
| Other | 77 (16.1) | 413 (83.9) | 1.0 | | | 1.0 | | |
| Hospital | 191 (23.5) | 621 (76.5) | 1.6 | 1.19 to 2.14 | 0.002 | 1.3 | 0.81 to 2.06 | 0.286 |
| **Type of health facility** | | | | | | | | |
| Public | 129 (23.2) | 426 (76.8) | 1.0 | | | 1.0 | | |
| Private not-for-profit | 55 (20.5) | 213 (79.5) | 0.9 | 0.60 to 1.22 | 0.380 | 0.8 | 0.51 to 1.27 | 0.353 |
| Private for-profit | 84 (18.0) | 382 (82.0) | 0.7 | 0.53 to 0.99 | 0.041 | 0.8 | 0.49 to 1.30 | 0.362 |
| **Region of the country** | | | | | | | | |
| Central | 148 (25.3) | 437 (74.7) | 1.0 | | | 1.0 | | |
| Eastern | 62 (15.1) | 348 (84.9) | 0.5 | 0.38 to 0.73 | <0.001 | 0.6 | 0.37 to 0.94 | 0.025 |
| Other | 58 (19.7) | 236 (80.3) | 0.7 | 0.52 to 1.02 | 0.066 | 0.8 | 0.50 to 1.22 | 0.270 |
| **Professional cadre** | | | | | | | | |
| Nurse | 111 (14.7) | 645 (85.3) | 1.0 | | | 1.0 | | |
| Non-nurse | 157 (29.5) | 376 (70.5) | 2.4 | 1.84 to 3.19 | <0.001 | 1.7 | 1.16 to 2.40 | 0.005 |
| **Age** | | | | | | | | |
| Less than 30 years | 119 (20.8) | 452 (79.2) | 1.0 | | | 1.0 | | |
| Aged 30 years or older | 149 (20.8) | 569 (70.3) | 1.0 | 0.76 to 1.30 | 0.969 | 0.9 | 0.65 to 1.31 | 0.647 |
| **Patient load** | | | | | | | | |
| Greater than 30/day | 128 (22.2) | 449 (77.8) | 1.0 | | | 1.0 | | |
| At most 30/day | 140 (19.7) | 572 (80.3) | 0.9 | 0.66 to 1.12 | 0.268 | 1.2 | 0.85 to 1.75 | 0.272 |
| **Department** | | | | | | | | |
| Surgery | 13 (13/1) | 86 (86.9) | 1.0 | | | 1.0 | | |
| Medicine | 150 (23.7) | 482 (76.3) | 2.1 | 1.12 to 3.79 | 0.021 | 2.1 | 0.99 to 4.38 | 0.054 |
| Paediatrics, Obs&Gyn | 40 (20.2) | 158 (79.8) | 1.7 | 0.85 to 3.30 | 0.136 | 2.0 | 0.90 to 4.57 | 0.090 |
| Other | 65 (18.1) | 295 (81.9) | 1.5 | 0.77 to 2.77 | 0.250 | 1.4 | 0.66 to 3.18 | 0.358 |
| **Involved in medical research** | | | | | | | | |
| No | 160 (17.6) | 749 (82.3) | 1.0 | | | 1.0 | | |
| Yes | 108 (38.6) | 272 (61.4) | 1.9 | 1.40 to 2.46 | <0.001 | 1.5 | 1.05 to 2.15 | 0.026 |
| **Ever encountered fatal ADR** | | | | | | | | |
| No | 197 (19.0) | 842 (81.0) | 1.0 | | | 1.0 | | |
| Yes | 71 (28.4) | 179 (71.6) | 1.7 | 1.24 to 2.32 | 0.001 | 1.1 | 0.71 to 1.64 | 0.732 |
| **Knowing to whom to report** | | | | | | | | |
| No | 129 (20.2) | 511 (79.8) | 1.0 | | | 1.0 | | |
| Yes | 139 (21.4) | 510 (78.6) | 1.1 | 0.82 to 1.41 | 0.577 | 1.2 | 0.86 to 1.74 | 0.254 |
| **Suggestions for improved ADR reporting** | | | | | | | | |
| No | 54 (17.0) | 264 (83.0) | 1.0 | | | 1.0 | | |
| Yes | 214 (22.0) | 757 (78.0) | 1.4 | 0.99 to 1.92 | 0.054 | 0.9 | 0.60 to 1.37 | 0.628 |
| **Received patient ADR complaint in past 4 weeks** | | | | | | | | |
| No | 73 ( 7.5) | 900 (92.5) | 1.0 | | | 1.0 | | |
| Yes | 195 (61.7) | 121 (38.3) | 19.9 | 14.3 to 27.6 | <0.001 | 19.0 | 13.5 to 27.1 | <0.001 |

ADR, adverse drug reaction; Obs&Gyn, Obstetrics and Gynaecology.

be done if there was certainty that it was related to the use of a particular drug'; OR=0.6, 95% CI 0.41 to 0.89) and lethargy ('I do not know how information reported in ADR form is used'), see table 6.

### Suggestions for improved ADR reporting

The most frequently cited suggestion was to sensitise, train and provide ongoing medical education on ADRs to HCPs (42%, 667/1589 suggestions) followed by making ADR forms available (17%, 262/1589), sensitising the public and counselling patients about ADRs (11%, 166/1589), creating a coordinating office in each health facility (5%, 73/1589), providing financial

incentives to reporters (4%, 65/1589) and making available telephone or online ADR reporting systems (4%, 57/1589), see table 7.

### DISCUSSION

A low proportion of HCPs reported having submitted an ADR report in the previous 12 months (15%) and the level of awareness of PV was also low, similar to observations made elsewhere.[34 42 43] HCPs from different cadres may recognise suspected ADRs but fail to take the responsibility to report.[44] Barely one in eight (13%) of suspected ADRs in the past month was reported by the

**Table 5** Personal and professional factors associated with ADR reporting in the past 12 months among 1164 healthcare professionals who had been in post for at least 1 year, Uganda, 2013

| Factor | ADR reporter | | Crude analysis | | | Adjusted analysis | | |
|---|---|---|---|---|---|---|---|---|
| | Yes (%) | N (%) | OR | 95% CI | p Value | OR | 95% CI | p Value |
| Level of health facility | | | | | | | | |
| Other | 36 (8.0) | 413 (92.0) | 1.0 | | | 1.0 | | |
| Hospital | 128 (17.9) | 587 (82.1) | 2.5 | 1.69 to 3.70 | <0.001 | 1.9 | 1.18 to 3.10 | 0.008 |
| Type of health facility | | | | | | | | |
| Public | 91 (18.5) | 402 (81.5) | 1.0 | | | 1.0 | | |
| Private not-for-profit | 40 (16.8) | 198 (83.2) | 0.9 | 0.59 to 1.34 | 0.585 | 0.8 | 0.50 to 1.23 | 0.286 |
| Private for-profit | 33 (7.6) | 400 (92.4) | 0.4 | 0.24 to 0.56 | <0.001 | 0.5 | 0.28 to 0.77 | 0.003 |
| Region of the country | | | | | | | | |
| Central | 82 (15.9) | 433 (84.1) | 1.0 | | | 1.0 | | |
| Eastern | 36 (9.7) | 334 (90.3) | 0.6 | 0.38 to 0.86 | 0.008 | 0.7 | 0.43 to 1.13 | 0.140 |
| Other | 46 (16.5) | 233 (83.5) | 1.0 | 0.70 to 1.55 | 0.836 | 1.2 | 0.75 to 1.84 | 0.471 |
| Professional cadre | | | | | | | | |
| Nurse | 93 (13.5) | 597 (86.5) | 1.0 | | | 1.0 | | |
| Non-nurse | 71 (15.0) | 403 (85.0) | 1.1 | 0.81 to 1.58 | 0.470 | 0.8 | 0.55 to 1.18 | 0.264 |
| Age | | | | | | | | |
| Less than 30 years | 70 (15.0) | 396 (85.0) | 1.0 | | | 1.0 | | |
| Aged 30 years or older | 94 (13.5) | 604 (86.5) | 0.9 | 0.63 to 1.23 | 0.455 | 0.6 | 0.43 to 0.91 | 0.014 |
| Patient load | | | | | | | | |
| Greater than 30/day | 84 (16.1) | 439 (83.9) | 1.0 | | | 1.0 | | |
| At most 30/day | 80 (12.5) | 561 (87.5) | 0.7 | 0.54 to 1.04 | 0.081 | 0.9 | 0.61 to 1.27 | 0.510 |
| Department | | | | | | | | |
| Surgery | 10 (11.5) | 77 (88.5) | 1.0 | | | 1.0 | | |
| Medicine | 95 (16.3) | 488 (83.7) | 1.5 | 0.75 to 3.00 | 0.253 | 2.3 | 1.08 to 4.73 | 0.030 |
| Paediatrics, Obs&Gyn | 18 (10.5) | 153 (89.5) | 0.9 | 0.40 to 2.06 | 0.065 | 0.8 | 0.36 to 1.95 | 0.675 |
| Other | 41 (12.7) | 282 (87.3) | 1.1 | 0.54 to 2.34 | 0.147 | 1.6 | 0.73 to 3.50 | 0.243 |
| Involved in medical research | | | | | | | | |
| No | 103 (12.6) | 716 (87.4) | 1.0 | | | 1.0 | | |
| Yes | 61 (17.7) | 284 (82.3) | 1.5 | 1.06 to 2.11 | 0.023 | 1.3 | 0.88 to 1.87 | 0.191 |
| Ever encountered fatal ADR | | | | | | | | |
| No | 98 (10.7) | 820 (89.3) | 1.0 | | | 1.0 | | |
| Yes | 62 (27.1) | 167 (72.9) | 3.0 | 2.12 to 4.33 | <0.001 | 2.9 | 1.94 to 4.25 | <0.001 |
| Knowing to whom to report | | | | | | | | |
| No | 62 (11.0) | 504 (89.1) | 1.0 | | | 1.0 | | |
| Yes | 102 (17.1) | 496 (82.9) | 1.7 | 1.19 to 2.35 | 0.003 | 1.7 | 1.18 to 2.46 | 0.005 |
| Suggestions for improved ADR reporting | | | | | | | | |
| No | 32 (10.6) | 270 (89.4) | 1.0 | | | 1.0 | | |
| Yes | 132 (15.3) | 730 (84.7) | 1.5 | 1.01 to 2.30 | 0.044 | 1.6 | 1.04 to 2.49 | 0.032 |

ADR, adverse drug reaction; Obs&Gyn, Obstetrics and Gynaecology.

HCPs in that same period, yet around three-fifths of patient ADR-complaints in the past month were adjudged by HCPs to be suspected ADRs. Integration of PV into pre-service training curricula and emphasising its importance in promoting patient safety in healthcare delivery is a first step[45 46] on which other PV initiatives can build.

To raise the number of submitted ADR reports, Uganda has proposed mandatory reporting of ADRs by industry and HCPs.[22] However, questions have been raised about the effectiveness of compulsory reporting by HCPs[47] and the NPC needs to improve its feedback to ADR reporters since our respondents ranked it much lower than ATIC. Moreover, HCPs in our study reported ADRs to a greater extent than in nationally reported statistics: 2% of HCPs (27/1281:95% CI 1.3% to 2.9%) had reported any suspected ADR to the NPC in the previous

year compared with the NPC's annual average national ADR reporting rate for Uganda from 2007 to mid-2013 of 0.44% (based on 1348 reports in 6.5 years from 46 566 clinical staff countrywide: 95% CI 0.38% to 0.51%) or 0.90% in the highest report-year of 2012 (413 reports in 2012:95% CI 0.80% to 0.97%). Thus, HCPs in our study seemed at least twice as likely to have submitted suspected ADRs to the NPC in the previous year when compared with the national ADR reporting rates by Uganda's HCPs.

One limitation to our estimates is that more than one HCP may have described (and reported) the same suspected ADR since our ability to discriminate between suspected ADRs was compromised by variation in the quality of ADR descriptions, a limitation that the NPC also contends with.

**Table 6** Attitudinal factors associated with adverse drug reaction (ADR) reporting in past 12 months among 1114 healthcare professionals who responded to attitudinal questions, Uganda, 2013

| Factor | Reported an ADR in the past 12 months | | Crude analysis | | | Adjusted analysis* | | |
|---|---|---|---|---|---|---|---|---|
| | Yes (%) | No (%) | OR | 95% CI | p Value | OR | 95% CI | p Value |
| I do not know how information reported in an ADR form is used | | | | | | | | |
| Agree | 64 (12.5) | 447 (87.5) | 0.7 | 0.47 to 0.97 | 0.031 | 0.7 | 0.46 to 1.00 | 0.052 |
| Neutral | 17 (10.6) | 143 (89.4) | 0.6 | 0.32 to 0.98 | 0.041 | 0.5 | 0.27 to 0.94 | 0.030 |
| Disagree | 81 (17.5) | 383 (82.5) | 1.0 | | | 1.0 | | |
| I would only report an ADR if I were sure that it was related to the use of a particular drug | | | | | | | | |
| Agree | 86 (12.2) | 620 (87.8) | 0.6 | 0.39 to 0.81 | 0.002 | 0.6 | 0.41 to 0.89 | 0.011 |
| Neutral | 12 (9.9) | 109 (90.1) | 0.4 | 0.23 to 0.87 | 0.015 | 0.6 | 0.29 to 1.17 | 0.128 |
| Disagree | 60 (19.7) | 244 (80.3) | 1.0 | | | 1.0 | | |

*Adjusted for personal and professional characteristics: level of health facility, type of health facility, region, non-nurse as professional cadre, age, patient load, department, involvement in medical research, ever encountered a fatal ADR, knowing to whom to report ADRs and suggesting ways to improve ADR reporting.

Consistent with ADR reports from the NPC,[17] we identified antibiotics, antiretroviral agents and antimalarials as the three most frequently cited medication classes in survey-described ADRs. Therefore, health initiatives already focusing on the PV of these medications, if replicated for other classes, present opportunities to strengthen overall PV systems in these settings.[17] As a PV exemplar in Uganda, the NPC and AIDS Control Programme introduced TSR in 2011 to monitor tenofovir for renal toxicity and to detect suspected ADRs related to antiretroviral therapy use in the Prevention of Mother to Child Transmission of HIV and in the Early Infants Diagnosis programme.[48] Results from TSR are yet to be disseminated, however.

Around three-fifths of patients' ADR-complaints to HCPs in the past month translated into ADR suspicion. Patient ADR-complaint was dominant among explanatory factors for HCPs' ADR suspicion in the past month and so we suggest that empowering patients to support HCPs may improve the detection and reporting of suspected ADRs. Moreover, other countries have instituted systems that promote spontaneous direct patient reporting of suspected ADRs, thus permitting patients to participate in PV activities that teach them to better handle their medicines and improve their communication with HCPs.[49 50]

Improvement of the ADR reporting form for Uganda seems necessary. Therefore, our research team designed a form that is relevant to the inpatient setting and captures additional information required for causality assessment of suspected medicines. This form will be tested in a follow-up study on inpatients.

Other suggestions to improve ADR reporting by respondents included: increased visibility of the NPC and giving useful feedback to ADR reporters, introducing telephone and online reporting systems, increasing onsite support supervision, making ADR forms more available, providing training and continued medical education of HCPs as suggested elsewhere,[51] and sensitising the public to ADRs. The absence of a national PV policy, however, coupled with the lack of proper coordination between the NPC and numerous health programmes and sentinel sites may undermine efforts to strengthen the countrywide PV system.[17] For example, in Uganda's

**Table 7** Suggested methods of improving adverse drug reaction (ADR) reporting among healthcare professionals, Uganda, 2013

| Method | Freqency | Per cent |
|---|---|---|
| Sensitise, train and give continuous medical education to healthcare professionals | 666 | 42.0 |
| Make forms available, eg, in patient hospital files in wards | 262 | 16.5 |
| Sensitise the public through media, posters and counsel patients about ADRs | 159 | 10.5 |
| Create liaison office to coordinate ADR reporting in each health facility | 74 | 4.6 |
| Incentivise reporting/motivate health workers/provide financial support | 65 | 4.1 |
| Provide toll-free telephone line or online ADR reporting system | 58 | 3.6 |
| Increase and strengthen onsite support/supervision | 38 | 2.4 |
| Compulsory ADR reporting | 23 | 1.4 |
| Give feedback to ADR reporters | 21 | 1.3 |
| Increase awareness of existence of the National Pharmacovigilance Centre | 21 | 1.3 |
| Other | 202 | 13.0 |
| Total | 1589 | 100 |

teaching hospitals, could some clinical grand rounds address PV and suspected serious ADRs?

Although previous studies suggested a positive relationship between older age and ADR reporting,[52 53] we found that older HCPs (≥30 years) were less likely than their younger counterparts to have reported suspected ADRs in the past 12 months. These contrasting results might be attributed to idiosyncratic differences between HCPs and healthcare systems in Europe and Africa in such a way that younger staff, as in our study, may have had more PV training. There is, as yet, limited published literature from other African settings. Our respondents were, on average, 10 years younger when compared with studies conducted in Europe.[29] We suggest that older HCPs in Uganda be targeted in future strategies on improved ADR reporting.

In contrast to other studies,[53] training on how to report ADRs was not significantly associated with increased ADR reporting. Given the cross-sectional study design we used, it was not possible to establish whether PV training preceded ADR reporting, or vice versa; therefore we were unable to assess their temporal relationship. That notwithstanding, Lopez-Gonzalez et al[8] have suggested that multifaceted interventions, as opposed to single educational programmes, increase to a greater extent HCPs' PV awareness and motivate them to report ADRs.

A low level of PV awareness may lead to under-reporting of ADRs.[54] In our study, knowing to whom to report was an important factor for ADR reporting in the final logistic regression. We also observed that the proportion (31%: 95% CI 29% to 34%) of respondents aware of the existence of Uganda's NPC is lower than reported for Nigeria (52% (51/99):95% CI 42% to 61%).[34] Much higher proportions of PV awareness have been reported in Europe[29] and Asia,[55 56] where there are higher ADR reporting rates per million of population[57] and more government involvement in national PV programmes.[34]

HCPs who had ever encountered a fatal ADR were twice as likely to report an ADR as HCPs who had not. Correspondingly, development of a serious or fatal ADR was the most frequently cited reason for ADR reporting. We also found that HCPs who suggested possible ways of improving the ADR reporting system were more likely to have reported an ADR in the previous 12 months.[58]

HCPs who agreed with the statement 'I would only report an ADR if I was sure that it was related to the use of a particular drug' (diffidence) were less likely to report suspected ADRs. Apart from diffidence and lethargy/indifference ('I do not know how information reported in the ADR form is used'), none of the other Inman factors was associated with ADR reporting.[8 32 59] Diffidence and lethargy can be targeted in educational interventions to promote ADR reporting and by improved feedback to ADR reporters.

Although provision of financial incentives to reporters was the fifth most frequently cited suggestion to improve ADR reporting, it was not statistically significant in the logistic regression for the odds on ADR reporting and these findings are consistent with those in the developed world.[60]

In private for-profit health facilities, HCPs were less likely to have reported ADRs in the previous 12 months than their counterparts in the public sector. In addition, HCPs in hospitals (public and private) were twice as likely as those from other health facilities (HCs II and III, community pharmacies, drug shops) to have reported suspected ADRs in the previous 12 months. Whereas few PV scale-up activities in Africa have given priority to the private sector,[16 22] more public–private collaboration could strengthen PV systems in our SSA setting.[61]

Our study had several limitations. First, we used self-report as the main method of enquiry and this may have introduced recall bias. Second, we may have experienced social desirability bias as HCPs may not have given frank responses for fear of being embarrassed if they were not reporting ADRs. However, as we used self-administered questionnaires without respondents' names, the potential for this bias was reduced. Third, the cross-sectional design that we used could not establish temporal relationships between ADR reporting in the past year and some explanatory factors. Fourth, there was over-representation of doctors and pharmacists/pharmacy technicians versus nurses. Finally, several respondents may have referred to the same suspected ADR but this did not have a significant bearing since our main focus was assessment of individual ADR reporting behaviour rather than on individual ADRs.

Our study has, however, generated key insights on determinants in Uganda for HCPs' ADR suspicion and reporting.

## CONCLUSIONS

One in five HCPs had suspected an ADR in the past 4 weeks while one in seven had reported an ADR in the previous 12 months. Empowering patients to support HCPs in suspected ADR detection and reporting is essential to strengthening PV systems in Africa. HCPs who had ever encountered fatal ADRs are keener reporters and can consequently help others to avoid the experience that made them better reporters. HCPs ought to know that they do not have to be certain about causality to report suspected ADRs. Poor access to suspected ADR forms and lack of feedback on reports are constraints that can be rectified.

**Author affiliations**
[1]Department of Pharmacology and Therapeutics, Makerere University College of Health Sciences, Kampala, Uganda
[2]Clinical Epidemiology Unit, Department of Medicine, Makerere University College of Health Sciences, Kampala, Uganda
[3]National Pharmacovigilance Centre, National Drug Authority, Kampala, Uganda
[4]Medical Research Council Biostatistics Unit, Cambridge, UK

**Acknowledgements** The authors wish to thank all the HCPs who agreed to participate in this study. They also thank Huldah Nassali at the National Pharmacovigilance Centre for providing technical support to this project. Finally, RK thanks Noeline Nakasujja at Makerere University College of Health Sciences, and Yukari C Manabe at Johns Hopkins University for supporting his development of the first three drafts of the manuscript.

**Contributors** RK conceived the study and drafted the manuscript and, along with SMB, participated in its design, implementation, statistical analysis and the drawing of inferences. CK, PW and HBN participated in study design and in the process of manuscript writing. All authors approved the final manuscript.

**Funding** This work was supported by Training Health Researchers into Vocational Excellence (THRiVE) in East Africa, grant number 087540, funded by the Wellcome Trust; grant number 5R24TW008886 supported by OGAC, NIH and HRSA; and an African Doctoral Dissertation Research Fellowship award (ADDRF Award 2013 - 2015 ADF 006.) offered by the African Population and Health Research Centre (APHRC) in partnership with the International Development Research Centre (IDRC). SMB holds GSK shares and is funded by Medical Research Council programme number MC_U105260794.

**Competing interests** None.

**Ethics approval** Ethical approval was obtained from the School of Medicine Research and Ethics Committee, Makerere University College of Health Sciences and the Uganda National Council for Science and Technology.

**Provenance and peer review** Not commissioned; externally peer reviewed.

**Data sharing statement** Categorical data are available from the lead author, RK, by email request to kiguba@gmail.com.

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
