## [Reviewer comments · BMJ Open]

Some articles will have been accepted based in part or entirely on reviews undertaken for other BMJ Group journals. These will be reproduced where possible.

ARTICLE DETAILS

TITLE (PROVISIONAL)	Recognition and reporting of suspected adverse drug reactions by surveyed healthcare professionals in Uganda: key determinants
AUTHORS	Kiguba, Ronlad; Karamagi, Charles; Waako, Paul; Ndagije, Helen; Bird, Sheila

VERSION 1 - REVIEW

REVIEWER	Dr. M. Ramesh JSS University, College of Pharmacy, India
REVIEW RETURNED	20-Jun-2014

GENERAL COMMENTS	Abstract requires to be re written to improve the clarity especially the presentation of study data. Pilot study should have been conducted to validate the questionnaire Detailed methodology should have been provided especially on site selection criteria, development and administration of questionnaire
--

REVIEWER	Maxine Gossell-Williams Univeristy of the West Indies, Dept of Basic Medical Sciences, Mona Campus/
REVIEW RETURNED	28-Jun-2014

GENERAL COMMENTS	The theoretical frame work was poorly presented. Authors showed little review of previous reports from Uganda (Bukirwa et al. Tropical medicine & international health TM & IH, 2008 Sept., v. 13, no. 9, p. 1143-1152. 13 9). Some use of word incorectly and words not known eg.'silloed'. Authors need help with presentation of their discriptive data. The paper has issues with sampling that may not reflect the true state. It does not add any new finding t the literature, howevr were it no for sampling issues probably BMJ would allow correction.
--

REVIEWER	Marion Bennie Strathclyde Institute of Pharmacy and Biomedical Sciences University of Strathclyde Glasgow Scotland
REVIEW RETURNED	10-Jul-2014

GENERAL COMMENTS	Minor observation - page 11 line 25 and 32 and table 2 - use of mean and SD is questionable where the sd is greater than the mean suggesting a non normal distribution - justification or amendment to median and IQR should be considered This is a well written report in an area of research where little has been documented on how to improve PV activity in SSA
--

VERSION 1 – AUTHOR RESPONSE

Reviewer Name Dr. M. Ramesh

Institution and Country JSS University, College of Pharmacy, India

Please state any competing interests or state 'None declared':None declared

B1. Abstract requires to be re written to improve the clarity especially the presentation of study data. To clarify, we have added a statement on ADR report-rate to the conclusions to reflect that ADR report-rate was one of the main objectives of the study. The sentence now reads: "One in five HCPs suspected an ADR in the past-month and one in seven reported an ADR in the previous year".

The results section of the Abstract is otherwise laid out logically as follows: the first paragraph reports on the basics (awareness of pharmacovigilance, the extent of ADR-suspicion and the extent of ADR-reporting); second paragraph reports on the factors associated with ADR suspicion; and the third on factors associated with ADR reporting. A statement on ADR reporting has been added to the conclusion section of the abstract.

B2. Pilot study should have been conducted to validate the questionnaire

An explanation on piloting is added as follows: "Prior to its administration, the questionnaire was elaborated by members of the research team who have diverse expertise in pharmacy, pharmacovigilance, and questionnaire design. Completion-time was tested by research assistants. Thereafter, an integrated pilot was conducted on 125 healthcare professionals. The subsequent revisions were sufficiently minor that results of the pre-test were included in the final analysis" - Methods section (page 10).

B3. Detailed methodology should have been provided especially on site selection criteria, development and administration of of questionnaire

Please see the answers above (B2) for information on the development and testing of the questionnaire.

Study sites were purposively selected. Details are provided on page 8 of the Methods section: "From 25 May 2012 through 28 February 2013, we conducted a survey across Uganda in purposively selected, geographically diverse public and private health facilities. Public institutions included the National Referral Hospital-Mulago, and six Regional Referral Hospitals each selected to represent a major region of the country. In addition, we included District Hospitals and Health Centres (HCs) at levels II to IV in the catchment area where a Regional Referral Hospital was selected. For logistical reasons, we selected a convenience sample of private for-profit and private not-for-profit health

facilities (which included drug shops) in the respective districts where public institutions were assessed. Permission to conduct the research was sought from the administrators of the selected institutions.”

“Any HCP involved in prescribing, transcribing, dispensing medication orders, and administration of drugs to a patient was eligible for inclusion. Written informed consent was obtained from HCPs prior to their recruitment. The self-completed questionnaires did not contain identifying information on individual HCPs. The survey team used serial numbers to track distributed questionnaires.”

Reviewer Name Maxine Gossell-Williams

Institution and Country Univeristy of the West Indies, Dept of Basic Medical Sciences, Mona Campus/
Please state any competing interests or state 'None declared': None to declare

C1. The theoretical frame work was poorly presented.

The study did not use a theoretical framework but employed a conceptual framework of several potential determinants/explanatory variables - mainly derived from prior literature from developed countries, please see introduction - that were regressed separately on the two outcomes, ADR suspicion and ADR reporting, to identify those that were significantly associated with those outcomes.

C2. Authors showed little review of previous reports from Uganda (Bukirwa et al. Tropical Medicine & International Health TM & IH, 2008 Sept., v. 13, no. 9, p. 1143-1152. 13 9).

This paper has now been cited. We thank the referee for sharing this qualitative research paper which, indeed, highlights the need for active participation of patients and healthcare professionals in voluntary reporting of adverse drug reactions.

C3. Some use of word incorectly and words not known eg.'siloed'.

The word 'siloed' has been dropped from the sentence and the word 'specific' adopted (Paragraph 2 line 6 under the title "Background").

The sentence now reads: "Strengthening of PV systems in sub-Saharan African (SSA) countries has received support from global health initiatives, but reporting is often disease specific (e.g. malaria, vaccines, HIV/AIDS) because of restricted funding streams rather than strengthening countrywide reporting systems".

C4. Authors need help with presentation of their discriptive data.

For all descriptive statistics, every effort has been made to present the data as fractions (numerator/denominator) and as percentages with their 95% confidence intervals, making it easier for the reader to assess precision of point estimates and to be aware of the typically low rate of non-response to specific questions. Citation of percentages without numerator/denominator incorrectly conceals these details. Tables and figures have also been presented as is statistically appropriate, please see Statistical Guidelines for Contributors to Medical Journals, which one of us [Bird, nee Gore] co-authored.

C5. The paper has issues with sampling that may not reflect the true state.

Limitations in sampling have been acknowledged but representation of healthcare professionals from all regions of the country enhances generalizability of the findings.

C6. It does not add any new finding t the literature, howevr were it no for sampling issues probably BMJ would allow correction.

To our knowledge, this is the largest survey on pharmacovigilance conducted among healthcare professionals in sub-saharan Africa. New findings include: documentation of the patient ADR-complaint rate, ADR-suspicion rate by HCPs, ADR-reporting rate by HCPs, and the determinants of the main outcomes (ADR-suspicion and ADR-reporting).

We are pleased that referee D acknowledges and welcomes this major addition to the literature from sub-saharan Africa.

We have, of course, acknowledged both a) lack of sampling frames and consequent practical difficulties for random sampling per centre and b) on-purpose, rather than at-random, achievement of geographical coverage. However, it is important not to make the best the enemy of the good. By describing our difficulties and limitations, we hope that others will seek to improve on our methods. The availability of our successfully-designed questionnaire allows them to focus on better documentation of response-rates and on achieving high overall willingness by eligible sites for the survey to take place in their location.

Reviewer Name Marion Bennie

Institution and Country Strathclyde Institute of Pharmacy and Biomedical Sciences

University of Strathclyde Glasgow Scotland

Please state any competing interests or state 'None declared':None declared

D1. Minor observation - page 11 line 25 and 32 and table 2 - use of mean and SD is questionable where the sd is greater than the mean suggesting a non normal distribution - justification or amendment to median and IQR should be considered

The reported means are based on large n (at least 100) and so the distribution of each mean will be approximately normally distributed despite the distribution of the number of ADR-complaints received or ADRs suspected being highly skewed (as $SD \gg \text{mean signals}$).

The reported means (and SD) of 3.5 (SD 9.5) for patient ADR-complaints and 2.2 (SD 2.6) for suspected ADRs on page 11 were thus purposely reported because the means are used to calculate the total number of patient ADR-complaints and the total number of suspected ADRs. The aim then was to determine the ADR suspicion rate per patient ADR complaint which the reader would find difficult to follow if we had reported the median values and we should have had difficulties in providing 95% CI for rate had we not been able to invoke central limit theorem re normality of means.

D2. This is a well written report in an area of research where little has been documented on how to improve PV activity in SSA

Thank you!